# Pharmacists’ Seasonal Influenza Vaccine Recommendations

**DOI:** 10.3390/pharmacy10030051

**Published:** 2022-04-25

**Authors:** Roland Langer, Mirjam Thanner

**Affiliations:** 1Department of Medical Sciences, Private University in the Principality of Liechtenstein, 9495 Triesen, Liechtenstein; 2Kantonsspital St. Gallen, 9007 St. Gallen, Switzerland; mirjam.thanner@kssg.ch

**Keywords:** influenza, pharmacists, recommendations, vaccine, healthcare worker, vaccine education, elderly patients, pregnancy, chronic disease

## Abstract

Seasonal influenza vaccination rates among European countries remain low despite the World Health Organization’s recommendations to vaccinate high-risk groups. Healthcare worker recommendations are strong predictors of increased vaccination uptake in the population. Therefore, this study aimed to analyze seasonal influenza vaccination recommendation behavior among pharmacists towards high-risk groups including patients, coworkers, and pharmacists’ family members during the COVID-19 pandemic. This cross-sectional, questionnaire-based research was conducted in Switzerland during the flu season and sent to all members of the Swiss Pharmacist Association. In December 2020, 569 community pharmacists completed the online survey. The influenza vaccination recommendation rates for high-risk patients were 93.6% for the elderly, 70.7% for pregnant women, 65.2% for immunocompromised people, and 60.3% for patients with chronic diseases. Pharmacists tend to recommend influenza immunization to patients more than to family members and colleagues. Holding a certification to administer immunization and personal influenza vaccine history were the main predictors for recommending influenza vaccination to patients, family members, and colleagues. Our results indicated that influenza vaccination recommendation rates in our whole sample of pharmacists, were higher for vaccinated and immunizing pharmacists. Ensuring high vaccinations rates and high ratio of immunizing pharmacists may be important in promoting seasonal influenza vaccination in the general population.

## 1. Introduction

Seasonal influenza causes annual disease outbreaks and poses a significant health risk. Influenza-associated respiratory disease has been reported to be attributable to approximately 291,243 to 645,832 annual deaths worldwide, most of which occur in the elderly (aged ≥65 years) [1]. Pregnant, immunocompromised, and chronically ill women have an increased risk of hospitalization and complications [2]. The Swiss Federal Office of Public Health and the World Health Organization recommend seasonal influenza vaccination (SIV) for these susceptible populations [3,4]. Flu vaccination is an effective strategy for reducing influenza-related hospitalization and mortality [5]. However, the actual SIV rates in Switzerland remained low, declining from approximately 16.4% in 2007 to 14.1% in 2012. SIV for high-risk groups was 38.5% for people aged ≥65 years, 5.1% for pregnant women, and 20–50% for patients suffering from chronic diseases. Other European countries have reported similar SIV rates [6,7].

Improving SIV rates during the COVID-19 pandemic is particularly important, as Influenza and COVID-19 share many symptoms [8]. This strategy was crucial at the time of the study, as no COVID-19 vaccine was available in Switzerland [9].

Flu vaccination in pharmacies has been reported to improve its uptake [10]. Several countries worldwide, have authorized their pharmacists to deliver vaccines; however, since the COVID-19 pandemic, many countries have recently changed their stance [11]. Pharmacist-based immunization programs are estimated to have the potential to affect >600 million individuals worldwide [12]. In 2015, the Swiss government widened the scope of community pharmacy practice to include various vaccine administrations, while COVID-19 immunization was added after in February 2021 [13,14]. Community pharmacists (CPs) receive certification to administer vaccines after completing a 4-day training course that covers theory and practice. Every two years, CPs must fulfill a continuing education obligation to maintain their certification.

Professional health recommendations are another effective way to improve SIV uptake. Recommendations from a healthcare worker (HCW) have been reported to have a great impact on vaccination rates, achieving success rates of >80% [15,16,17,18,19,20]. Patients visit a community pharmacy approximately 10 more times than their primary care physicians, allowing CPs several opportunities to act as educators by providing accurate information and recommendations regarding vaccines [21].

Only a few studies have focused on SIV recommendations rates, and most of these studies assessed physicians recommendations behavior, ranging from 0% in Asian countries to 95.9% in Western European countries [20,22,23,24,25,26,27,28].

This study aimed to estimate Swiss pharmacists’ SIV recommendation rates in the general population during the COVID-19 pandemic and to identify possible demographic factors associated with providing recommendations. In addition, this research helped to gain further insight into CPs willingness to offer COVID-19 vaccination in their pharmacies.

## 2. Materials and Methods

### 2.1. Study Design and Participants

This questionnaire-based research was conducted during the seasonal influenza season of 2020–2021, among CPs. On 1 December 2020, the Swiss Association of Pharmacists (Pharmasuisse, Bern-Liebefeld, Switzerland) distributed an invitation link to all its members, approximately 5900 pharmacists.

Before commencing the voluntary survey, participants were informed regarding the objectives and methodology of the study. The participants were asked to complete the survey in 3 weeks, and no incentives were offered to them. Confidentiality and anonymity of the gathered information were maintained, and no data identifying a participant were collected. All data were gathered using an online survey software (SoSci Survey, Munich, Germany).

### 2.2. Development and Calibration of the Questionnaire

An anonymous questionnaire was designed and adapted to the specificities of the Swiss Health System after reviewing existing survey tools from previous empirical research [29,30,31,32]. The feasibility of the survey in French and German was evaluated in a pilot study of several CPs. Based on the feedback, minor changes were incorporated to improve the final questionnaire. The data obtained from these CPs were excluded in the analysis. Only complete questionnaires were included in the final results and then translated into English.

The survey had several questions related to sociodemographics, certification to administer vaccines, personal SIV status in the previous season, and SIV recommendation behavior. Participants who recommended SIV only to the high-risk group could then select a single or multiple subgroup. Additionally, in relation to the COVID-19 pandemic, CPs were asked whether they would offer COVID-19 vaccination to patients once available. Variables related to willingness to offer COVID-19 vaccination were evaluated on a 5-point Likert scale, which was dichotomized into “likely” (including “likely” and “very likely”), “undecided”, and “unlikely” (including “unlikely” and “very unlikely”).

### 2.3. Data Analysis

All data analyses were performed using IBM SPSS, version 26.0 (IBM Corp., 2019, Armonk, NY, USA). Only fully completed questionnaires were used for the analyses. To determine significant associations, logistic regression was used to calculate odds ratios (ORs) with 95% confidence intervals. Statistical tests were considered significant when *p* values were <0.05.

### 2.4. Ethics Approval

Our project was voluntary and used fully anonymized data; thus, ethical approval for this study was waived by the ethics committee of Zurich.

## 3. Results

### 3.1. Demographics

The questionnaire was sent to approximately 5900 pharmacists, of whom 569 completed the questionnaire (response rate ≈ 9.64%). Five questionnaires were excluded, as responses were provided by pharmacists that were not working in community pharmacy settings. The final sample included 564 CPs (Table 1). The majority of the remaining participants were authorized to administer injections (*n* = 374, 66.3%).

### 3.2. Pharmacist Characteristics

Pharmacist demographics are reported in Table 1. Most pharmacists were female (413, 72.5%), worked in an urban setting (487, 86.3%), and were aged >40 years (346, 61.4%). The majority of respondents worked in medium-sized teams of 3 to 5 pharmacists, 389 (69.0%) were authorized to administer injections (*n* = 374, 66.3%), and almost half (48%) received SIV in the 2019/2020 season.

### 3.3. Certification

Among 192 non-certified pharmacists, 53 (32.8%) indicated that they were planning to become certified, 70 (36.5%) indicated that they were not planning to become certified, and 59 (30.7%) were unsure. Those who were planning to become certified were significantly more likely to be female, aged <40 years, and practicing in a team with fewer pharmacists. Most pharmacists received certification in 2016 (*n* = 74, 19.9%), 2017 (*n* = 101, 27.2%), and 2018 (*n* = 89, 23.9%).

### 3.4. COVID-19 Vaccine

The majority (314, 84.4%) of certified pharmacists were willing to offer COVID-19 vaccination to patients if they were legally allowed, once a vaccine would be available. Of the pharmacists, 98 (26.3%) were undecided, and 13 (0.8%) were against offering COVID-19 vaccinations.

### 3.5. SIV Recommendations to High-Risk Groups

Figure 1 summarizes the frequency of CP recommendations for high-risk groups. The recommendation rates for the elderly, immunocompromised patients, patients suffering from chronic diseases, and pregnant women were 93.6%, 70.7%, 65.2%, and 60.3%, respectively. For each subgroup, patients received recommendations more frequently than pharmacist’s family members or colleagues.

### 3.6. SIV Recommendations to Family Members

Overall, 23.2% of the CPs recommended SIV to their own family regardless of belonging to a high-risk group, while 51.2% recommended it to at least 1 subgroup. In the multivariate analysis (Table 2), having a certification to vaccinate and personal SIV status were associated with a higher probability of recommending SIV for each group. A rural practice area was a positive predictor for recommending SIV to people with chronic disease and to the elderly. However, CPs aged <30 years were less likely to recommend SIV to immunocompromised family members and to family members suffering from chronic diseases.

### 3.7. SIV Recommendations to Patients

Of all the CPs, almost one-third (33.2%) recommended SIV to patients regardless of their risk status, and 62.1% recommended it to at least 1 vulnerable group. Multivariate regression analysis (Table 3) revealed that certification to administer the vaccine was significantly associated with recommendation uptake in all subgroups, while personal SIV history was associated with recommendations for pregnant women, immunocompromised people, and people with chronic diseases.

### 3.8. SIV Recommendations to Coworkers

Of all the CPs, 32.3% recommended SIV to colleagues regardless of them belonging to a high-risk group, while 27.7% recommended SIV to coworkers belonging to a subgroup. Table 4 presents the factors that significantly contributed to the SIV recommendations according to the multivariate regression analyses. Pharmacists’ past vaccination status and holding a certification to administer vaccines was significantly associated with all risk groups, while CPs aged ≥50 years were more likely to recommend SIV to all subgroups, except to pregnant colleagues. Hierarchy played a role in some subgroups; pharmacy owners and managers were more willing to recommend the vaccine to some of the subgroups.

## 4. Discussion

To the best of our knowledge, this study is the first to generate important insights into Swiss CPs’ SIV recommendations. A wide range of results have been reported in similar studies conducted in other countries for other healthcare professionals [20,22,23,24,25,26,27,28]. An overall recommendation similar to other European countries, for example, 60.1% of CPs recommended SIV to pregnant patients, whereas the European average for different HCWs was 58.9% [25]. Gynecologists in East Germany have recommended SIV to the elderly, with a slightly higher rate of 95.9% [28]. In Spain, pharmacists recommended SIV more frequently to people with chronic disorders (94.8%) and slightly less frequently to immunocompromised patients (63.7%) [32].

In 2012, Swiss SIV rates for people aged 65+ years, pregnant patients, and patients suffering from chronic diseases were 38.5%, 5.1%, and 20–50%, respectively [6]. These vaccination rates were significantly lower than the recommendation rates in our study. The discrepancy between the number of immunizations and the recommendation reported by HCWs is consistent with the literature and could be due to the fact that not all healthcare professionals that recommend SIV are offering vaccination in everyday practice [22,33,34].

Although we did not investigate the reasons for recommending SIV, other studies have identified that the main reasons were national guidelines, relevant training on SIV, and personal vaccination status. Being aware of national guidelines and the influenza vaccine priority group increased the likelihood of recommending SIVs [22,23]. Other studies have revealed that improving HCWs’ knowledge of SIV and influenza may increase the odds of recommending SIV [20,23,27,35]. These results suggest the importance of educating HCWs on the scientific importance of the vaccine and prevention of the disease. These two predictors were addressed during a rigorous immunization certification training program. In addition to practice-based activities, CPs are trained on vaccine-preventable diseases, vaccines, and national vaccination guidelines. This could explain why CPs with certification to administer vaccines were more likely to recommend SIV. Common findings in previous studies have indicated that HCW SIV history was a strong predictor of SIV recommendations in patients [20,22,23,25,27,31]. However, our study did not completely converge with these findings, as this was a common predictor of recommendations in all subgroups excepts for patients aged ≥65 years.

Barriers to the SIV recommendations should also be addressed. The main predictors addressed in the literature regarding the willingness to recommend SIV were safety concerns, lack of awareness of national guidelines, and cost issues. Concerns on side effects and lower vaccine confidence have been associated with lower recommendation rates [23,25,28,36]. In previous studies, a lack of awareness of national guidelines was associated with a lower likelihood of recommending SIV for pregnant women [22,34]. These topics are addressed during the immunization certification training program and might explain why non-certified CPs deliver fewer SIV recommendations. The lack of adequate reimbursement by insurance and recommendation interviews being too time-consuming were cited by various HCWs [27,28,34]. In Switzerland, only SIV at the physician’s office are reimbursed, as opposed to the British model, where pharmacy-based SIV for high-risk groups are reimbursed by health insurance [37,38]. This might also be a reason why CPs in our study recommended SIV to a lesser extent in pregnant patients, immunocompromised patients, and patients suffering from chronic diseases. In Switzerland, CPs are not allowed to vaccinate these higher-risk groups, with a few exceptions, and they will not be able to make a financial gain from that recommendation. CPs may also be more hesitant to make recommendations to these patients because it may be more a physician’s role in providing the vaccine. Similar findings were observed among primary physicians and obstetrician–gynecologists when recommending SIV vaccines to pregnant women [25].

To conduct effective recommendation conversation, CPs should be aware of patients’ reasons for refusing SIV to address them accordingly. The common barriers to SIV reported in the literature are underestimated risk of severe influenza, lack of knowledge of having an indication for influenza vaccination, fear of side effects, and cost related to vaccination [17,18,19]. This is particularly true, as SIV performed in pharmacies is not reimbursed for most patients.

If vaccinated and certified CPs are more likely to recommend vaccines to patients, families, and coworkers, increasing the proportion of certified and vaccinated CPs is important. Hence, Swiss universities started to offer immunization certification programs to pharmacy students during the last year of their masters. Future studies could also investigate what leads current CPs to be certified or vaccinated and assess the barriers, as 36.5% reported unwillingness to attend a certification course, and 30.7% were still unsure.

Receiving a recommendation by any provider is often cited as a strong predictor of immunization, and overcoming vaccination hesitancy may decrease the inhibition threshold for future years [15,16,17,18,19,20,39]. Although CPs are not legally allowed to administer SIV to all higher-risk groups, they may encourage patients to receive the vaccine from a different provider, which may likely increase the vaccination rates of the general population in a snowball system. The uptake of influenza vaccination was observed when routine SIV recommendations were implemented in conjunction with increased access to vaccination [19].

This study had some important limitations. First, this was a cross-sectional study, which only reflects the current recommendation behavior and does not describe changes over time. Second, the participation rate was low, and we cannot exclude the possibility that the CPs that participated had a higher interest in the subject and different recommendation practices than those who refused to participate. Third, SIV recommendation behavior during the COVID-19 pandemic might be higher as interest in SIV was higher during the pandemic and might not reflect practices during non-pandemic years [40]. This may have affected the generalization of the results, and the recommendation rates may have been overestimated. Fourth, self-reported behaviors may be susceptible to recall bias and could result in the overestimation of positive recommendation behavior; however, the survey was conducted during the seasonal influenza season, which could limit this effect. Lastly, the findings are not representative of all Swiss CPs, as we included only those that had membership in the Swiss Pharmacist Association.

Nonetheless, our study has some strengths, such as the use of a nationwide web survey available in German and French. Moreover, to the best of our knowledge, this is the first study to assess recommendation behaviors among Swiss CPs.

## 5. Conclusions

A personal vaccination history and certification to administer immunization among CPs may contribute to higher SIV recommendations for patients, family members, and colleagues. The elderly had the highest recommendation rates, as these were the primary risk groups that CPs were allowed to vaccinate. To increase CPs’ compliance to actively recommend SIV, regulators should implement an appropriate vaccination course even for non-immunizing CPs and focus on SIV campaigns towards pharmacists. Future work should include measuring the impact of CP recommendations on vaccination rates.

## Figures and Tables

**Figure 1 pharmacy-10-00051-f001:**
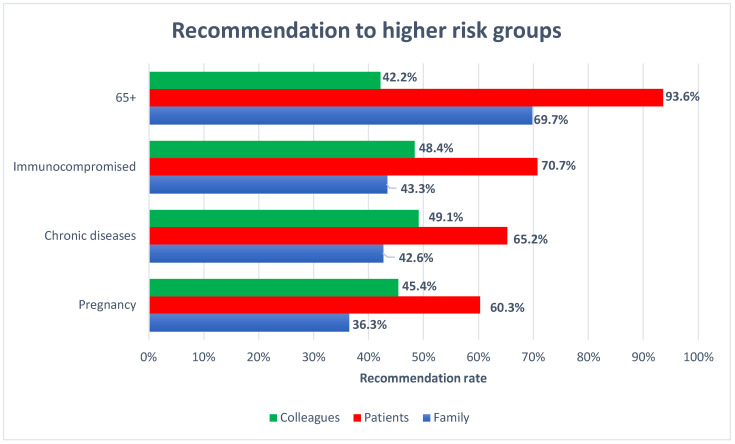
CPs’ recommendation overview.

**Table 1 pharmacy-10-00051-t001:** Characteristics of Survey Respondents.

Characteristics	N	%
**Sex**		
Female	409	72.5%
Male	155	27.5%
**Age**		
<30	74	13.1%
30–39	144	25.5%
40–49	173	30.7%
50+	173	30.7%
**Certification**		
Yes	372	66.0%
No	192	34.0%
**Practice area**		
Rural	77	13.7%
Urban	487	86.3%
**Position**		
Owner	93	16.5%
Manager	152	27.0%
Assistant manager	227	40.2%
Non-managerial	92	16.3%
**CP per team**		
1 to 2	153	27.1%
3 to 5	389	69.0%
6+	22	3.9%
**SIV season 2019/2020**		
Yes	271	48.0%
No	293	52.0%

**Table 2 pharmacy-10-00051-t002:** Season influenza vaccination recommendations to family members.

	Pregnant Women		Chronic Disease		Immunocompromised		Aged 65+ Years	
	OR	95% CI	OR	95% CI	OR	95% CI	OR	95% CI
**Sex**								
Female	1.14	0.74–1.77	0.78	0.51–1.19	0.80	0.52–1.22	0.81	0.52–1.25
Male	–	–	–	–	–	–	–	–
**Age**								
<30	0.52	0.26–1.05	0.42 *	0.22–0.80	0.49 *	0.26–0.92	1.22	0.63–2.35
30–39	1.40	0.84–2.31	0.94	0.57–1.54	0.89	0.54–1.46	0.98	0.59–1.64
40–49	1.01	0.63–1.62	0.92	0.58–1.46	0.69	0.43–1.09	1.45	0.88–2.37
50+	–	–	–	–	–	–	–	–
**Certification**								
Yes	2.02 *	1.33–3.08	2.15 *	1.44–3.23	2.17 *	1.45–3.25	1.85 *	1.24–2.76
No	–	–	–	–	–	–	–	–
**Practice area**								
Rural	1.48	0.87–2.52	2.38 *	1.39–4.07	0.64	0.38–1.10	3.09 *	1.55–6.17
Urban	–	–	–	–	–	–	–	–
**Position**								
Owner	0.99	0.52–1.88	0.76	0.40–1.43	0.75	0.40–1.42	0.75	0.37–1.50
Manager	1.18	0.66–2.11	0.80	0.45–1.42	0.94	0.53–1.67	0.74	0.40–1.38
Assistant manager	0.55	0.32–0.94	0.62	0.36–1.05	0.71	0.42–1.21	0.67	0.38–1.18
Non-managerial	–	–	–	–	–	–	–	–
**SIV status 2019/20**								
Yes	2.80 *	1.92–4.09	2.79 *	1.94–4.02	3.28 *	2.27–4.72	2.24 *	1.51–3.33
No	–	–	–	–	–	–	–	–
**CP per team**								
1 to 2	0.55	0.20–1.48	0.58	0.22–1.55	0.87	0.32–2.34	1.60	0.59–4.33
3 to 5	0.84	0.33–2.15	0.94	0.37–2.39	1.24	0.48–3.21	1.21	0.48–3.10
6+	–	–	–	–	–	–	–	–

* *p* < 0.05.

**Table 3 pharmacy-10-00051-t003:** Season influenza vaccination recommendations to patients.

	Pregnant Women		Chronic Disease		Immunocompromised		Aged 65+ Years	
	OR	95% CI	OR	95% CI	OR	95% CI	OR	95% CI
**Sex**								
Female	1.35	0.90–2.04	1.25	0.83–1.89	0.85	0.55–1.32	0.45	0.19–1.16
Male	–	–	–	–	–	–	–	–
**Age**								
<30	0.93	0.51–1.71	0.96	0.51–1.81	0.70	0.36–1.35	1.31	0.32–5.27
30–39	0.77	0.47–1.26	0.76	0.46–1.25	0.70	0.41–1.20	0.40	0.16–0.97
40–49	1.05	0.66–1.67	1.02	0.63–1.63	0.55	0.34–0.89	1.21	0.45–3.29
50+	–	–	–	–	-	–	–	–
**Certification**								
Yes		1.28–2.76	1.95 *	1.33–2.87	1.52 *	1.02–2.26	2.43 *	1.17–5.05
No	–	–	–	–	–	–	–	–
**Practice area**								
Rural	0.58	0.34–1.01	0.47	0.26–0.85	0.79	0.45–1.40	0.33	0.08–1.44
Urban	–	–	–	–	–	–	–	–
**Position**								
Owner	1.24	0.66–2.34	0.97	0.51–1.85	1.13	0.58–2.17	1.69	0.45–6.30
Manager	1.52	0.86–2.71	1.10	0.62–1.95	1.39	0.77–2.49	1.33	0.44–4.01
Assistant manager	0.79	0.47–1.32	1.09	0.65–1.86	1.34	0.79–2.28	0.95	0.37–2.45
Non-managerial	–	–	–	–	–	–	–	–
**SIV status 2019/20**								
Yes	1.80 *	1.25–2.58	1.78 *	1.23–2.59	1.72 *	1.17–2.53	0.59	0.29–1.22
No	–	–	–	–	–	–	–	–
**CP per team**								
1 to 2	0.66	0.25–1.74	1.17	0.45–3.05	1.10	0.40–2.98	0.50	0.06–4.59
3 to 5	1.05	0.42–2.66	1.32	0.53–3.29	1.07	0.42–2.77	0.50	0.06–4.14
6+	–	–	–	–	–	–	–	–

* *p* < 0.05.

**Table 4 pharmacy-10-00051-t004:** Season influenza vaccination recommendations to colleagues.

	Pregnant Women		Chronic Disease		Immunocompromised		Aged 65+ Years	
	OR	95% CI	OR	95% CI	OR	95% CI	OR	95% CI
**Sex**								
Female	1.36	0.90–2.06	1.12	0.74–1.68	1.06	0.70–1.60	1.20	0.79–1.83
Male	–	–	–	–	–	–	–	–
**Age**								
<30	0.92	0.50–1.69	0.36 *	0.19–0.67	0.47 *	0.25–0.86	0.41 *	0.22–0.77
30–39	0.88	0.54–1.43	0.57 *	0.35–0.93	0.56 *	0.34–0.91	0.52 *	0.32–0.85
40–49	0.75	0.48–1.18	0.51 *	0.32–0.80	0.37 *	0.24–0.59	0.36 *	0.23–0.58
50+	–	–	–	–	-	–	–	–
**Certification**								
Yes	2.10 *	1.42–3.11	2.42 *	1.64–3.58	1.80 *	1.22–2.65	2.02 *	1.35–3.01
No	–	–	–	–	–	–	–	–
**Practice area**								
Rural	0.65	0.39–1.09	0.90	0.54–1.50	0.89	0.53–1.48	0.71	0.42–1.19
Urban	–	–	–	–	–	–	–	–
**Position**								
Owner	2.41 *	1.28–4.52	1.91 *	1.01–3.62	1.95 *	1.03–3.68	1.32	0.70–2.48
Manager	1.63	0.93–2.84	2.13 *	1.21–3.77	2.68 *	1.51–4.76	1.73	0.98–3.07
Assistant manager	0.77	0.46–1.29	1.31	0.78–2.20	1.40	0.83–2.37	0.90	0.53–1.53
Non-managerial	–	–	–	–	–	–	–	–
**SIV status 2019/20**								
Yes	1.56 *	1.08–2.23	1.57 *	1.09–2.24	1.94 *	1.35–2.77	1.92 *	1.34–2.76
No	–	–	–	–	–	–	–	–
**CP per team**								
1 to 2	0.57	0.22–1.49	1.51	0.58–3.92	1.16	0.45–3.03	1.02	0.39–2.70
3 to 5	0.42	0.17–1.05	1.00	0.40–2.51	0.85	0.34–2.11	1.02	0.41–2.58
6+	–	–	–	–	–	–	–	–

* *p* < 0.05.

## Data Availability

The data presented in this study are available upon request from the corresponding author.

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
