# Peer review of "Pharmacists’ Seasonal Influenza Vaccine Recommendations"

_pharmacy, 2022, doi:10.3390/pharmacy10030051_

Round 1

Reviewer 1 Report

Thank you for work and contribution to the journal. I have some minor recommendations to improve the paper.

  1. In the abstract, It would be helpful to either delete, or mention earlier, that the pharmacist were asked about recommendations to the family/colleagues.  It is unclear if the question pertained to the pharmacists family or the patient's (at risk) family.
  2. In the introduction, may want to add whether or not pharmacist were certified to administer the COVID-19 vaccine (and that it was not available at the time of the study) as I was a little confused until I read later it was not yet in Switzerland.
  3. Did the participants receive any reminder about the survey in the 3 week time, if yes, please include in the methods.
  4. Consider changing 2.1 to Study Design and Participants, and move the first paragraph under "survey tools" to this section. Then move the second paragraph of the "Survey Tools" to the 2.3 Development and Calibration of the Questionnaire. It will help with flow (Survey Tools would then not be needed.
  5. The last paragraph (page 2, lines 91-95) are repeated with lines 78-81, can delete one of them.
  6. In the results, add a survey response rate (%)
  7. Page 4, Section 3.3, lines 139-140.  I am unclear as to the recommendations were made to the pharmacist's family member (the colleague part is clear) or the high risk patient's family member, please clarify in the methods.
  8. Page 5, lines 148 - do not need to capitalize Chronic
  9. In the Conclusions, I would delete the first sentence as the survey was not specifically designed to measure recommendations during a pandemic (unless there is a question regarding difference of practice during a pandemic and not during a pandemic. The willingness to administer a vaccine is important but not the main finding of the paper.  Can include it after the influenza findings.

Author Response

Thank you!

Reviewer 2 Report

Thank you for the opportunity to review the manuscript Pharmacists’ seasonal influenza vaccine recommendations. The title suggests an important and interesting paper.

Line 1: I note that page 1 refers to the manuscript as a Review but this appears to be original research.

Abstract

The abstract reads well and provides a concise summary of the research to come.

Line 11: the term “questionnaire-based survey” seems redundant. The terms questionnaire and survey are often interchangeable. Consider questionnaire-based research or survey-based research.

Line 22: please define SIV in abstract or avoid use of acronym here.

Introduction

The introduction provides a clear background on the topic of the research.

Line 29: this is a very precise number of deaths. What time period does this cover?

Lines 36-37: given the sentence before, it sounded like flu vaccine rates were low due to COVID but then you quote numbers from 2012. Perhaps disconnect the two sentences or provide more recent figures, if possible.

Lines 40-41: can you elaborate on which countries allow pharmacists to administer flu vaccines? I think there were many, as opposed to “Only a few”, even pre-COVID. And this seems incongruous with lines 43-44 when you quote the >600 mil individuals impacted by pharmacist-based immunization.

Lines 60-61: the text shifts into first person when had previously been written in third person. Consider consistency throughout.

Materials and Methods

Line 64: the term “questionnaire-based survey” seems redundant, as above.

Lines 91-94 appear to repeat the same information provided in lines 80-82.

Results

Table 1: check results for age 40-49 and 50+ as they report the same numbers in the table and don’t add up to the 315, 55.8% reported in text (lines 117-118).

Line 161: Table 2 is duplicated. Table 3 appears to be missing.

Discussion

Line 178: not sure what you mean by diffuse results in this context.

In the paragraph on cost issues, it might be interesting to compare the Swiss model to other countries. For example, Australian pharmacists can provide influenza vaccine in community pharmacies with the patients paying but high-risk patients can access the vaccine for free from doctors, but not pharmacists.

Author Response

Thank you!
